# Engineering circular RNA for potent and stable translation in eukaryotic cells

R. Alexander Wesselhoeft [1,2], Piotr S. Kowalski[3] & Daniel G. Anderson[1,3,4,5]

Messenger RNA (mRNA) has broad potential for application in biological systems. However, one fundamental limitation to its use is its relatively short half-life in biological systems. Here we develop exogenous circular RNA (circRNA) to extend the duration of protein expression from full-length RNA messages. First, we engineer a self-splicing intron to efficiently circularize a wide range of RNAs up to 5 kb in length in vitro by rationally designing ubiquitous accessory sequences that aid in splicing. We maximize translation of functional protein from these circRNAs in eukaryotic cells, and we find that engineered circRNA purified by high performance liquid chromatography displays exceptional protein production qualities in terms of both quantity of protein produced and stability of production. This study pioneers the use of exogenous circRNA for robust and stable protein expression in eukaryotic cells and demonstrates that circRNA is a promising alternative to linear mRNA.

[1] David H. Koch Institute for Integrative Cancer Research, Massachusetts Institute of Technology, Cambridge, MA 02139, USA. [2] Department of Biology, Massachusetts Institute of Technology, Cambridge, MA 02139, USA. [3] Department of Chemical Engineering, Massachusetts Institute of Technology, Cambridge, MA 02139, USA. [4] Institute for Medical Engineering and Science, Massachusetts Institute of Technology, Cambridge, MA 02139, USA. [5] Harvard and MIT Division of Health Science and Technology, Massachusetts Institute of Technology, Cambridge, MA 02139, USA. Correspondence and requests for materials should be addressed to D.G.A. (email: dgander@mit.edu)

Circular RNAs (circRNAs) endogenous to eukaryotic cells have drawn increasing interest due to their prevalence and range of potential biological functions[1]. Most circRNAs found in nature are generated through backsplicing[2–4] and appear to fulfill noncoding roles[1, 5, 6]. However, it has been demonstrated that some circRNAs endogenous to *Drosophila* and humans encode proteins[7, 8]. In addition to having protein-coding potential, endogenous circRNAs lack the free ends necessary for exonuclease-mediated degradation, rendering them resistant to several mechanisms of RNA turnover and granting them extended lifespans as compared to their linear mRNA counterparts[2, 9]. For this reason, circularization may allow for the stabilization of mRNAs that generally suffer from short half lives[10, 11] and may therefore improve the overall efficacy of exogenous mRNA in a variety of applications[12].

Previous efforts to increase mRNA stability include the use of untranslated regions such as those present in the native beta globin mRNA, methylguanosine cap analogs to protect mRNA from decapping enzymes, nucleoside modification, and codon optimization. These strategies have yielded modest improvements in RNA stability[10, 13–16], and thus alternative approaches to RNA stabilization, such as circularization, are desirable. However, the efficient circularization of long in vitro transcribed (IVT) RNA, the purification of circRNA, and the adequate expression of protein from circRNA are significant obstacles that must be overcome before their protein-coding potential can be realized. In this study, we present an engineering approach to generating exogenous circRNAs for potent and durable protein expression in eukaryotic cells.

## Results

**Long RNA circularization is assisted by homology**. There are three general strategies for exogenous RNA circularization: chemical methods using cyanogen bromide or a similar condensing agent, enzymatic methods using RNA or DNA ligases, and ribozymatic methods using self-splicing introns[17–19]. A ribozymatic method utilizing a permuted group I catalytic intron has been reported to be more applicable to long RNA circularization and requires only the addition of GTP and Mg2+ as cofactors[17]. This permuted intron-exon (PIE) splicing strategy consists of fused partial exons flanked by half-intron sequences[20]. In vitro, these constructs undergo the double transesterification reactions characteristic of group I catalytic introns, but because the exons are already fused they are excised as covalently 5′ to 3′ linked circles[17] (Fig. 1a). Using this strategy as a starting point for creating a protein coding circular RNA, we inserted a 1.1 kb sequence containing a full-length encephalomyocarditis virus (EMCV) IRES, a Gaussia luciferase (GLuc) message, and two short regions corresponding to exon fragments (E1 and E2) of the PIE construct between the 3′ and 5′ introns of the permuted group I catalytic intron in the thymidylate synthase (Td) gene of the T4 phage[21] (Fig. 1a, Supplementary Data 1). Precursor RNA was synthesized by run-off transcription and then heated in the presence of magnesium ions and GTP to promote circularization, essentially as described previously for the circularization of shorter RNAs[21]. However, we were unable to obtain splicing products. We speculated that long intervening regions between splice sites which may reduce the ability of the splice sites to interact with one another and form a stable complex, thus reducing splicing efficiency. Indeed, the intervening region between the 5′ and 3′ splice sites of native group I introns is on average 300–500 nucleotides long[22], while the intervening region of the engineered RNA that we constructed was two to four-fold longer. Therefore we designed perfectly complementary 'homology arms' 9 (weak) or 19 (strong) nucleotides in length placed at

the 5′ and 3′ ends of the precursor RNA with the aim of bringing the 5′ and 3′ splice sites into proximity of one another (Fig. 1b, Supplementary Data 1). Addition of these homology arms increased splicing efficiency from 0 to 16% for weak homology arms and to 48% for strong homology arms as assessed by disappearance of the precursor RNA band (Fig. 1c). To ensure that the major splicing product was circular, we treated the splicing reaction with RNase R (Fig. 1d, Supplementary Fig. 1a). Sequencing across the putative splice junction of RNase R-treated splicing reactions revealed ligated exons, and digestion of the RNase R-treated splicing reaction with oligonucleotide-targeted RNase H produced a single band in contrast to two bands yielded by RNase H-digested linear precursor (Fig. 1d, e, Supplementary Fig. 1a). These data show that circRNA is a major product of these splicing reactions and that agarose gel electrophoresis allows for simple and effective separation of circular splicing products from linear precursor molecules, nicked circles, splicing intermediates, and excised introns.

**Spacer sequences improve circularization efficiency**. In order to further improve the efficiency of circRNA generation from the self-splicing precursor RNA, we considered other factors that may influence successful circularization. The 3′ PIE splice site is proximal to the IRES, and because both sequences are highly structured, we hypothesized that sequences within the IRES may interfere with the folding of the splicing ribozyme, either proximally at the 3′ splice site or distally at the 5′ splice site through long-distance contacts. In order to allow these structures to fold independently, we designed a series of spacers between the 3′ PIE splice site and the IRES that we predicted would either permit or disrupt splicing (Fig. 2a, Supplementary Data 1). Permissive spacers were designed to conserve secondary structures present within intron sequences that may be important for ribozyme activity, while the disruptive spacer was designed to disrupt sequences in both intron halves, especially the 5′ half. The addition of spacer sequences predicted to permit splicing increased splicing efficiency from 46 to 87% (P1 and P2), while the addition of a disruptive spacer sequence completely abrogated splicing (Fig. 2b). This improved construct, containing both homology arms and rationally designed spacers, was able to circularize RNA approaching 5 kb in length (Supplementary Fig. 1b).

**Anabaena catalytic intron is superior for circularization**. We also explored the use of an alternative group I catalytic intron from the Anabaena pre-tRNA[20]. Here we applied the same optimization techniques that we used to increase the efficiency of the permuted T4 phage intron splicing reaction. Interestingly, during our optimizations we noted that switching from the T4 catalytic intron to the Anabaena catalytic intron may have resulted in the weakening of a short stretch of internal homology between the IRES and the 3′ end of the coding region, which may have aided in the formation of an isolated splicing bubble (Fig. 2c, Fig. 3a). Strengthening this internal homology further increased splicing efficiency from 84 to 95% using the permuted Anabaena catalytic intron (Fig. 2c, d, Supplementary Data 1). We also found that use of the Anabaena catalytic intron resulted in a 37% reduction in circRNA nicking compared to the T4 catalytic intron (Fig. 2d, e). Due to increased splicing efficiency and intact circRNA output, the engineered Anabaena PIE system proved to be overall superior to the engineered T4 PIE system (Fig. 2e, Supplementary Fig. 2a).

**Autocatalytic sequences are compatible with coding regions**. Internal homology between exon 2 and the GLuc coding sequence

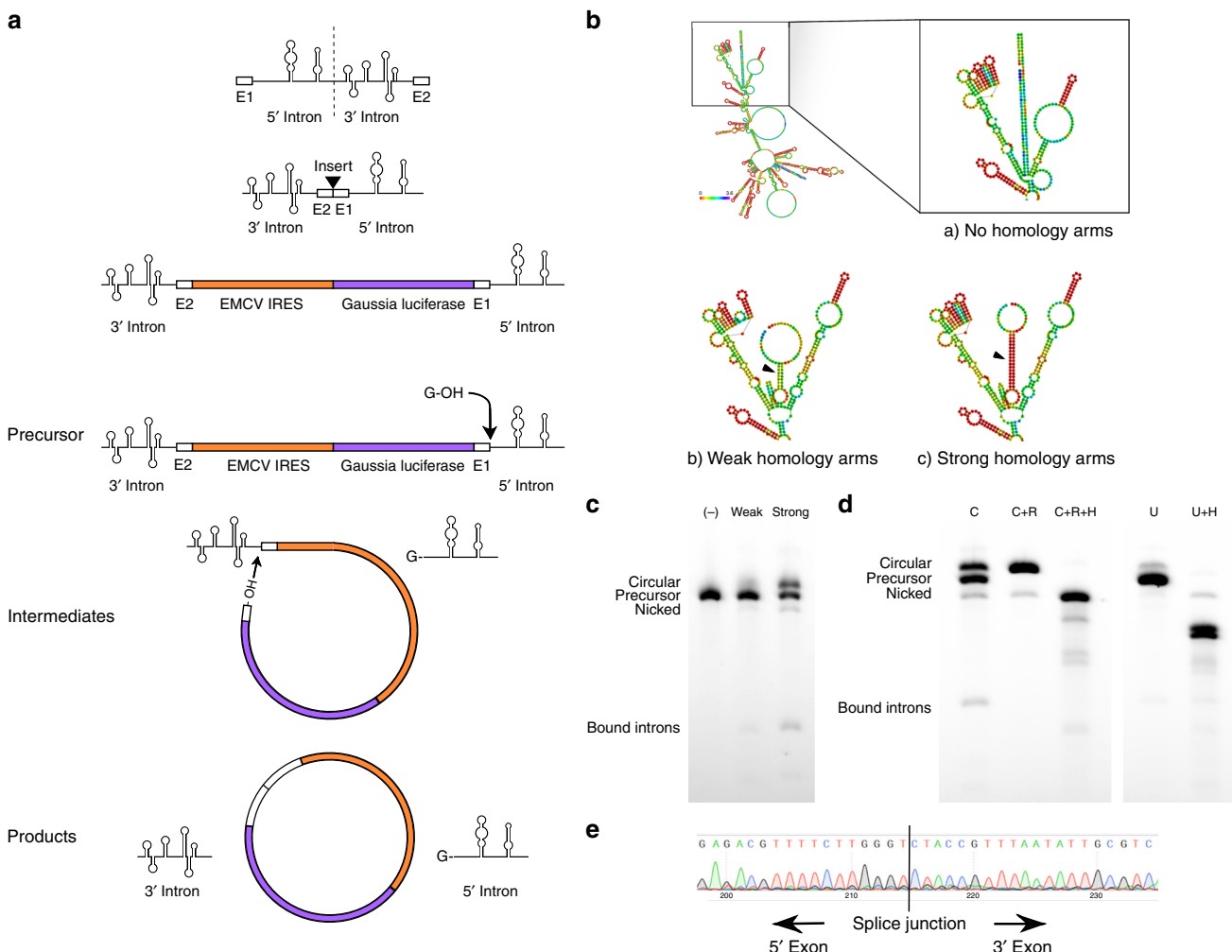

**Fig. 1** Permuted intron-exon splicing and addition of homology arms. **a** Schematic diagram showing permuted intron-exon construct design and mechanism of splicing. The group I catalytic intron of the T4 phage Td gene is bisected in such a way to preserve structural elements critical for ribozyme folding. Exon fragment 2 is then ligated upstream of exon fragment 1, and a coding region roughly 1.1 kb in length is inserted between the exon-exon junction. During splicing, the 3′ hydroxyl group of a guanosine nucleotide engages in a transesterification reaction at the 5′ splice site. The 5′ intron half is excised, and the freed hydroxyl group at the end of the intermediate engages in a second transesterification at the 3′ splice site, resulting in circularization of the intervening region and excision of the 3′ intron. **b** RNAFold predictions of precursor RNA secondary structure for homology arm design. Colors denote base pairing probability, with red indicating higher probability. Without homology arms, no base pairing is predicted to occur between the ends of the precursor molecule. **c** Agarose gel demonstrating the effect of homology arms on splicing. Putative circRNA runs at a higher molecular weight than heavier precursor RNA, as indicated. (−): no homology arms. Weak: weak homology arms, 9 nt. Strong: strong homology arms, 19 nt. **d** Agarose gel confirmation of precursor RNA circularization. C: precursor RNA (with strong homology arms) subjected to circularization conditions. C + R: Lane C, digested with RNase R. C + R + H: Lane C + R, digested with oligonucleotide-guided RNase H. U: precursor RNA not subjected to circularization conditions. U + H: Lane U, digested with oligonucleotide-guided RNase H. **e** Sanger sequencing output of RT-PCR across the splice junction of the sample depicted in lane C + R from (**d**)

rendered the optimized Anabaena PIE system incompatible with non-GLuc intervening regions. To adapt our circRNA construct for efficient circularization of a variety of long intervening RNA sequences, we de novo designed a pair of spacer sequences based on our understanding of the parameters that affect permuted catalytic group I intron splicing efficacy. These spacer sequences were engineered with three priorities: (1) to be unstructured and non-homologous to the proximal intron and IRES sequences; (2) to separate intron and IRES secondary structures so that each element is able to fold and function independently of one another; and (3) to contain a region of spacer–spacer complementarity to promote the formation of a sheltered splicing bubble containing catalytic intronic sequences flanked by two regions of homology (Fig. 3a, Supplementary Data 1). We also included homology arms at the 5′ and 3′ ends of the precursor molecule. Between these sequences we inserted an EMCV IRES, as well as coding regions for five different proteins, including Gaussia luciferase (total length: 1289nt), Firefly luciferase (2384nt), eGFP (1451nt), human erythropoietin (1313nt), and Cas9 endonuclease (4934nt). We were able to achieve circularization of all five RNA sequences (Fig. 3b, Supplementary Data 1); circularization efficiency matched that of our stepwise-designed construct (Fig. 2e) and was highly reproducible between inserts but was also dependent on size, with long RNAs less efficiently circularized (Supplementary Fig. 3a). In addition we found that long circRNAs were more prone to nicking in the presence of magnesium ions, resulting in accumulation of nicked circRNA during and after in vitro transcription and RNase R digestion

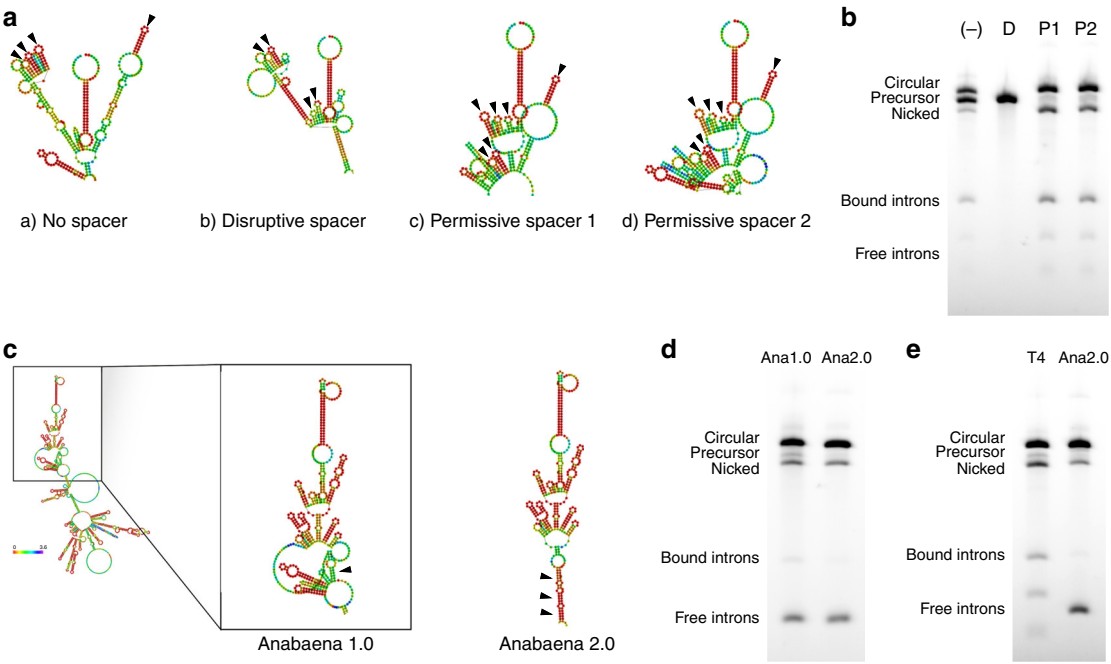

**Fig. 2** Spacer design and splicing using the Anabaena autocatalytic intron. **a** RNAFold predictions of precursor RNA secondary structure in the context of designed spacers. Secondary structures potentially important for ribozyme function are identified by black arrows. **b** Agarose gel demonstrating the effect of spacers on splicing. (−): no spacer. D: disruptive spacer. P1: permissive spacer 1. P2: permissive spacer 2. **c** RNAFold predictions of precursor RNA secondary structure for internal homology region design. Lack of significant internal homology (Anabaena 1.0) and introduced internal homology (Anabaena 2.0) indicated by black arrows. Splicing bubble indicated as the region between homology arms and internal homology regions that contains the splicing ribozyme. **d** Agarose gel demonstrating the effect of internal homology on splicing. **e** Agarose gel comparing the optimized T4 phage splicing reaction to the optimized Anabaena splicing reaction. Anabaena intron halves are of roughly equal lengths, and are less likely to remain associated after splicing in comparison to the T4 phage intron halves despite stronger homology arms

which reduced the overall yields and the purity of the RNase R-treated sample (Fig. 3b, c, Supplementary Fig. 3a). RNase R did not fully digest resistant Anabaena introns (Fig. 3b, bottom bands) or rare circular concatenations representing approximately 1.07% of splicing reactions (Fig. 3b, faint top bands).

**Exogenous circRNA is efficiently translated.** It has been demonstrated that endogenous circRNA may produce small quantities of protein[7]. As a means of assessing the ability of engineered circRNAs to produce protein, we transfected RNase R-digested splicing reactions of each construct into human embryonic kidney cells (HEK293). Transfection of Gaussia or Firefly luciferase circRNA resulted in robust production of functional protein as measured by luminescence (Fig. 3d, f). Likewise, we were able to detect human erythropoietin in cell culture media from transfection of erythropoietin circRNA, and observe EGFP fluorescence from transfection of EGFP circRNA (Fig. 3e, g). Co-transfection of Cas9 circRNA with sgRNA directed against GFP into HEK293 cells constitutively expressing GFP resulted in ablated fluorescence in up to 97% of cells in comparison to an sgRNA-only control (Fig. 3h, Supplementary Fig. 3d, e). Because RNase R digestion of splicing reactions is not always complete and precursor RNA contains a functional IRES, we also created a splice site deletion mutant of the GLuc construct to measure the potential contribution of impurities to protein expression. When transfected at equal weight quantities to RNase-R digested splicing reactions, this splice site deletion mutant produced a barely detectable level of protein (Supplementary Fig. 3b, c).

**The CVB3 IRES is superior in the context of circRNA.** To establish exogenous circRNA as a reliable alternative to existing

linear mRNA technology it is desirable to maximize protein expression. Cap-independent translation mediated by an IRES can exhibit varying levels of efficiency depending on cell context and is generally considered less efficient than cap-dependent translation when included in bicistronic linear mRNA[23]. Similarly, the polyA tail stabilizes and improves translation initiation efficiency in linear mRNA through the actions of polyadenylate binding proteins[24, 25]. However, the efficiency of different IRES sequences and the inclusion of a polyA tract within the context of circRNA has not been investigated. We replaced the EMCV IRES with 5′ UTR sequences from several viral transcripts that contain known or putative IRESs, as well as several other putative IRES sequences (Supplementary Data 1, Supplementary Fig. 4a)[26]. We found that the IRES from Coxsackievirus B3 (CVB3) was 1.5-fold more effective than the commonly adopted EMCV IRES in HEK293 cells (Fig. 4a). Because secondary structures proximal to the IRES, including within the coding region that directly follows the IRES, have the potential to disrupt IRES folding and translation initiation, we also tested selected viral IRES sequences in the context of Firefly luciferase. While the CVB3 IRES was still superior to all others, the efficacy of several other IRESs, most notably the Poliovirus IRES, was dramatically altered (Fig. 4a). We then tested whether the addition of an internal polyA sequence or a polyAC spacer control to IRES sequences that showed the ability to drive protein production above background levels from engineered circRNA would alter protein expression. We found that both sequences improved expression in all constructs, possibly due to greater unstructured separation between the beginning of the IRES sequence and the exon–exon splice junction, which is predicted to maintain a stable structure (Fig. 4b, Supplementary Fig. 4b). This greater degree of unstructured separation may reduce steric hindrance occluding

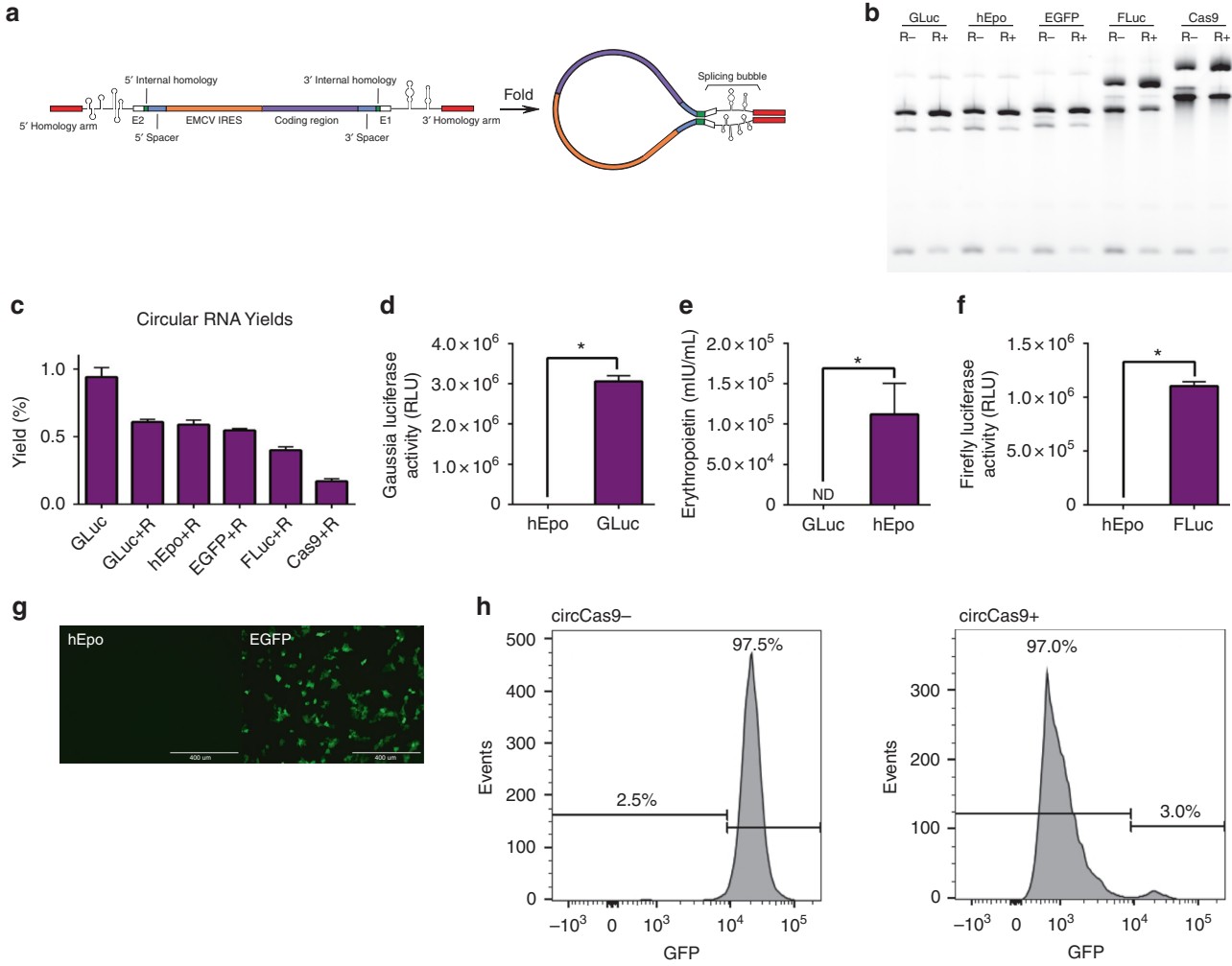

**Fig. 3** Evaluation of circularization efficacy and translation for a range of protein-coding circRNAs generated from de-novo engineered precursor RNA. **a** Schematic diagram showing elements of the engineered self-splicing precursor RNA design. **b** Agarose gel of precursor RNA containing an EMCV IRES and a variable insert including Gaussia luciferase (GLuc), human erythropoietin (hEpo), EGFP, Firefly luciferase (FLuc), or Cas9 coding regions after circularization and recircularization (R−). CircRNA was enriched by RNase R degradation (R+). **c** Approximate circRNA yields from treatment of 20 μg of splicing reaction with RNase R, as assessed by spectrophotometry ($n = 3$). **d** Luminescence in the supernatant of HEK293 cells 24 h after transfection with circRNA coding for GLuc ($n = 4$). **e** Expression of human erythropoietin in the supernatant of HEK293 cells 24 h after transfection with circRNA coding for hEpo as assessed by solid-phase sandwich ELISA ($n = 4$). **f** Luminescence in the lysate of HEK293 cells 24 h after transfection with circRNA coding for FLuc ($n = 4$). **g** GFP fluorescence in HEK293 cells 24 h after transfection with circRNA coding for EGFP (scale bar: 400 μm). **h** FACS analysis demonstrating GFP ablation in HEK293-EF1a-GFP cells 4 days after transfection with sgGFP alone (circCas9−) or co-transfected with circRNA coding for Cas9 (circCas9+), indicated by the appearance of a GFP-negative cell population (all data presented as mean + SD, *$p < 0.05$ (Welch's $t$-test))

initiation factor binding to IRES structures. In the case of EMCV and Poliovirus IRESs, polyA sequences improved expression beyond the improvement seen with an unstructured polyAC spacer. This may suggest that the association of polyadenylate binding proteins could enhance IRES efficiency. After selecting the most effective polyA or polyAC construct for each IRES, we explored IRES efficacy in different cell types, including human cervical adenocarcinoma (HeLa), human lung carcinoma (A549), and immortalized mouse pancreatic beta cells (Min6). We found that IRES efficacy varied depending on cell type, but the CVB3 IRES was superior in all types tested (Fig. 4c, d). Deletion of relatively short proximal or distal CVB3 IRES sequences resulted in dramatic loss of protein expression (Supplementary Fig. 4c).

**CircRNA can be purified from splicing reactions using HPLC.** Purity of circRNA preparations is another factor essential for maximizing protein production from circRNA and for avoiding

innate cellular immune responses. It has been shown that removal of dsRNA by HPLC eliminates immune activation and improves translation of linear nucleoside-modified IVT mRNA[16]. However, no scalable methods have been reported for purification of circRNA from byproducts of IVT and circularization reactions, which include dsRNA and triphosphate-RNA that may engage RNA sensors and induce a cellular immune response[16]. While the complete avoidance of nicked circRNA was untenable due to mild degradation during processing, we were able to obtain substantially pure (90% circular, 10% nicked) circRNA using gel extraction for small quantities and size exclusion HPLC for larger quantities of splicing reaction starting material (Fig. 5a, b). In both cases, we followed purification with RNase R treatment to eliminate the majority of degraded RNA. When comparing the protein expression of gel extracted or HPLC purified circRNA to RNase-R digested splicing reactions, we found that HPLC purification was a superior method of purification that surpassed RNAse R digestion alone (Fig. 5b, c).

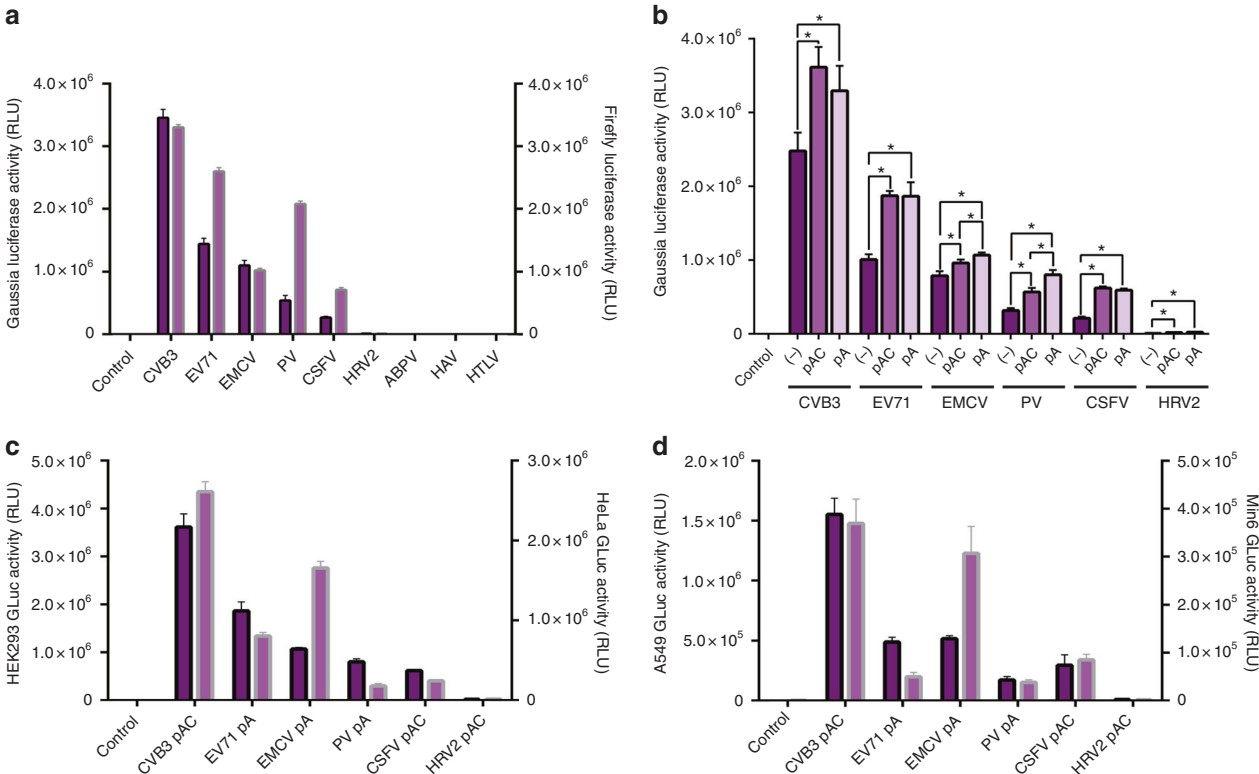

**Fig. 4** IRES efficacy in varying cell and sequence contexts. **a** Luminescence in the supernatant of HEK293 cells 24 h after transfection with circRNA containing a panel of viral 5′ UTR IRES sequences in GLuc (left axis, dark purple with black outline) and FLuc (right axis, light purple with gray outline) contexts. **b** Luminescence in the supernatant of HEK293 cells 24 h after transfection with circRNA containing an added polyA(30) or polyAC(30) spacer sequence separating the IRES from the splice junction. (−): no spacer. pAC: 30nt spacer consisting of adenosines and cytosines. pA: 30nt spacer consisting of adenosines. **c** Luminescence in the supernatant of HEK293 (left axis, dark purple with black outline) and HeLa (right axis, light purple with gray outline) cells 24 h after transfection with the most effective circRNAs by IRES in (**b**). **d** Luminescence in the supernatant of A549 (left axis, dark purple with black outline) and Min6 (right axis, light purple with gray outline) cells 24 h after transfection with the most effective circRNAs by IRES in (**b**) (all data presented as mean + SD, $n = 4$, *$p < 0.05$ (Welch's $t$-test))

**Protein production is comparable to linear mRNA**. It is unknown whether exogenous circRNA translation efficiency is comparable to that of linear mRNA, and whether circRNA protein production exhibits differences in stability. Using HPLC-purified engineered circRNA, we compared the stability and efficacy of Gaussia luciferase-coding circRNA (CVB3-GLuc-pAC) to equimolar quantities of a canonical unmodified 5′ methylguanosine-capped and 3′ polyA-tailed linear GLuc mRNA, as well as a commercially available nucleoside-modified (pseudouridine, 5-methylcytosine) linear GLuc mRNA (Trilink). Protein production assessed by luminescence 24 h post-transfection revealed that circRNA produced 811.2% more protein than the unmodified linear mRNA at this early time point in HEK293 cells (Fig. 6a). Interestingly, circRNA also produced 54.5% more protein than the modified mRNA, demonstrating that nucleoside modifications are not necessary for robust protein production from circRNA. Similar results were obtained in HeLa cells (Fig. 6a) and using optimized circRNA coding for human erythropoietin in comparison to linear mRNA modified with 5-methoxyuridine (Supplementary Fig. 5a, b). Luminescence data collected over 6 days showed that protein production from circRNA was extended relative to that from the linear mRNA in HEK293 cells, with circRNA exhibiting a protein production half-life of 80 h, while the half-lives of protein production from the unmodified and modified linear mRNAs were approximately 43 and 45 h, respectively (Fig. 6b). Due to increased expression or stability, circRNA also produced substantially more protein than both the unmodified and modified linear mRNAs over its

lifetime, while weaker initial expression did not abrogate the observed stability phenotype (Fig. 6c, Supplementary Fig. 6c). In HeLa cells, circRNA exhibited a protein production half-life of 116 h, while the half-lives of protein production from the unmodified and modified linear mRNA were approximately 44 and 49 h, respectively (Fig. 6d). This again resulted in substantially more protein production from circRNA over its lifetime compared to both the unmodified and modified linear mRNAs (Fig. 6c). We did not observe this enhanced expression or stability from a capped and polyadenylated circRNA precursor containing all accessory sequences (Supplementary Fig. 6a, b).

## Discussion

Obtaining stable protein production from exogenous mRNA has been a long-standing goal of mRNA biotechnology. Previous efforts to increase mRNA stability have yielded modest improvements[10, 13–16]. The possibility of adapting circular RNA for the purpose of stabilizing mRNA has been stifled by low circRNA production efficiency, difficulty of purification, and weak protein expression. Indeed, these obstacles must be overcome before the stability of protein production from circRNA can be fully assessed. Previously, the permuted group 1 catalytic intron-based system has been used to circularize a wide range of short RNA sequences in vitro, with circularization efficiencies reported to reach 90% for RNAs between 58 and 124 nt[20, 21]. Longer RNAs (up to 1.5 kb) have recently been circularized using this method, but circRNA yields from these constructs have not

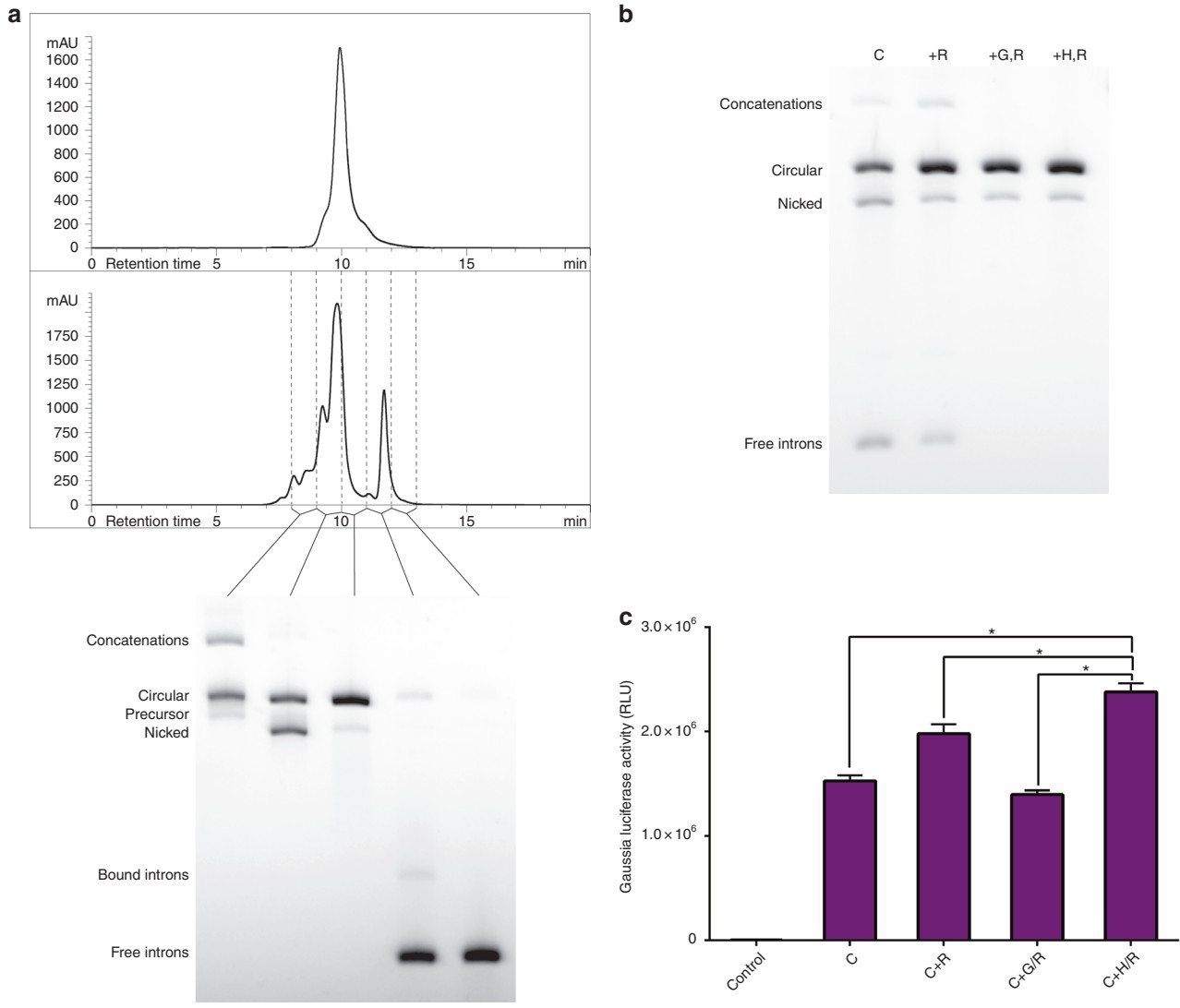

**Fig. 5** HPLC purification of circRNA from splicing reactions. **a** HPLC chromatogram of linear GLuc RNA (top) and a CVB3-GLuc-pAC splicing reaction (middle). Agarose gel of collected fractions (bottom). **b** Agarose gel of CVB3-GLuc-pAC purified by different methods. C: splicing reaction. +R: splicing reaction treated with RNase R. +G,R: splicing reaction gel extracted, and then treated with RNase R. +H,R: splicing reaction HPLC purified, and then treated with RNase R. **c** Luminescence in the supernatant of HEK293 cells 24 h after transfection with the CVB3-GLuc-pAC splicing reactions purified by different methods as noted in (**b**) (data presented as mean + SD, $n = 4$, *$p < 0.05$ (Welch's $t$-test))

been reported[28]. The engineered permuted group 1 catalytic intron-based system described herein permits the circularization of sequences up to 5 kb in length, significantly longer than previously reported. Moreover, we are able to obtain nearly 100% circularization efficiency for a range of inserted coding regions with diverse sequence compositions. We show that optimized circRNA is capable of producing large quantities of protein and also that it can be effectively be purified by HPLC. Finally, we demonstrate that circRNA can produce greater quantities of protein for a longer duration than the unmodified and modified linear RNAs used in this study, providing evidence that circRNA holds potential as an alternative to mRNA for the stable expression of protein. However, more work needs to be done to fully investigate the potential of circRNA for therapeutic and non-therapeutic applications, including further optimizations of protein translation from circRNA and a comprehensive study of the translation efficiency and stability of circRNA in other cell types and tissues. The nicking of longer circRNAs that we observed during the production of circRNA (Supplementary Fig. 2a), likely mediated by magnesium-catalyzed

autohydrolysis[27], significantly reduces yields and is another deficiency that requires improvement. Longer circRNAs have more opportunities to auto-hydrolyze simply based on their size. This degradation is more evident for circRNA than it is for linear RNA because single nicks in circRNA manifest as a single band on a gel, whereas single nicks in linear RNA manifest as a smear that is diluted over a range of molecular weights. This phenomenon is visible in Fig. 3b, lane 7 (FLuc, R-, as well as several other lanes); two major bands exist in this lane, with the top major band representing intact circRNA while the bottom major band represents nicked circRNA. No smear extends from the intact circRNA band, while a smear extends from the nicked circRNA band. The nicked circRNA band represents single nicks that occur at random positions in an intact circRNA, while the smear that extends from it represents additional nicks beyond the first that occur at random positions in the linearized nicked circRNA and thus give rise to degradation products of varying molecular weights. We do not expect circRNA nicking to present challenges beyond those that already exist for maintaining the stability of linear RNA in solution and during processing.

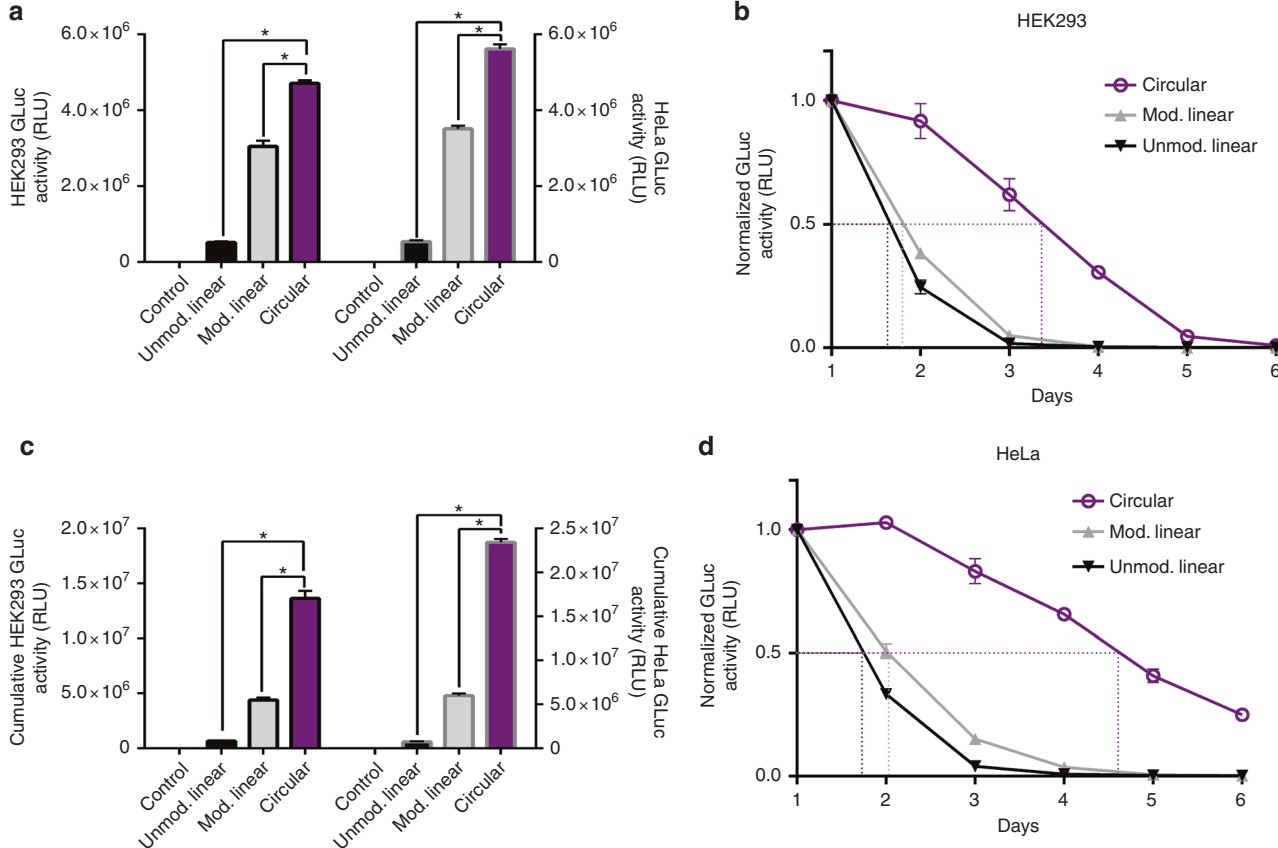

**Fig. 6** Translation efficacy of circRNA compared to linear mRNA. **a** Luminescence in the supernatant of HEK293 (left, black outline) and HeLa (right, gray outline) cells 24 h after transfection with CVB3-GLuc-pAC circRNA or modified or unmodified linear GLuc mRNA ($n = 4$ HEK293, $n = 3$ HeLa). **b** Luminescence in the supernatant of HEK293 cells starting 24 h after transfection with CVB3-GLuc-pAC circRNA or modified or unmodified linear GLuc mRNA and continuing for 6 days ($n = 4$). **c** Relative cumulative luminescence produced over 6 days by HEK293 (left, black outline) and HeLa (right, gray outline) cells transfected with CVB3-GLuc-pAC circRNA or modified or unmodified linear GLuc mRNA ($n = 4$ HEK293, $n = 3$ HeLa). **d** Luminescence in the supernatant of HeLa cells starting 24 h after transfection with CVB3-GLuc-pAC circRNA or modified or unmodified linear GLuc mRNA and continuing for 6 days ($n = 3$). (all data presented as mean + SD, *$p < 0.05$ (Welch's $t$-test))

It has recently been demonstrated that transfection of cells with exogenous circRNA results in the activation of antiviral gene products such as OAS, PKR, and RIG-I, and that RIG-I is likely to be the principal component responsible for the cellular response against circRNA[28]. We observed our circRNA preparations to elicit a significantly stronger IFN-α and IL-6 cytokine response in A549 cells than untransfected controls, although the response was comparable to that from unmodified linear mRNA (Supplementary Fig. 7a). We also found a 30-fold induction of IFN-β1 and a 3-fold induction of RIG-I transcripts upon transfection of HeLa cells with circRNA, compared to mock transfection (Supplementary Fig. 7b). The 3-fold induction of RIG-I mRNA we observed was significantly lower than the 500-fold induction reported by Chen et al. in HeLa cells[28] and was not sufficient to suppress protein translation and the stability of protein expression from circRNA in this cellular context (Fig. 6d). The observed differences in RIG-I activation could be attributed to differences in construct design and circRNA purification methods. To further suppress an immune response directed against circRNA, methods such as nucleoside modification could potentially be incorporated into circRNA design[13, 16]. Additional analysis of exogenous circRNA immunogenicity in the context of potential applications is required.

Finally, we show that circRNA is efficiently delivered to cells in vitro by a cationic lipid transfection reagent. However, a thorough investigation of suitable vehicles for the delivery of circRNA in vitro and in vivo is required.

## Methods

**Cloning and mutagenesis**. Protein coding, group I self-splicing intron, and IRES sequences were chemically synthesized (Integrated DNA Technologies) and cloned into a PCR-linearized plasmid vector containing a T7 RNA polymerase promoter by Gibson assembly using a NEBuilder HiFi DNA Assembly kit (New England Biolabs). Spacer regions, homology arms, and other minor alterations were introduced using a Q5 Site Directed Mutagenesis Kit (New England Biolabs). Primers for this and the following methods can be found in Supplementary Data 1.

**circRNA design and purification**. RNA structure was predicted using RNAFold[18]. Modified linear GLuc mRNA was obtained from Trilink Biotechnologies and consisted of a codon optimized GLuc coding region, a proprietary synthetic 5′ untranslated region, an alpha globin 3′ untranslated region, a cap 1 structure, a 120-nucleotide polyA tail, and complete replacement of uridine and cytosine along the entire mRNA with pseudouridine and 5-methylcytosine, respectively. Modified hEpo mRNA was also obtained from Trilink Biotechnologies and was structurally identical to the Trilink GLuc mRNA described above, except that it was modified with 5-methoxyuridine and the coding region coded for human erythropoietin. Unmodified linear RNA consisted of a GLuc or hEpo coding region but did not include specific untranslated regions. Unmodified linear mRNA or circRNA precursors were synthesized by in-vitro transcription from a linearized plasmid DNA template using a T7 High Yield RNA Synthesis Kit (New England Biolabs). After in vitro transcription, reactions were treated with DNase I (New England Biolabs) for 20 min. After DNase treatment, unmodified linear mRNA was column purified using a MEGAclear Transcription Clean-up kit (Ambion). RNA was then heated to 70 °C for 5 min and immediately placed on ice for 3 min, after which the RNA was capped using mRNA cap-2′-O-methyltransferase (NEB) and Vaccinia capping

enzyme (NEB) according to the manufacturer's instructions. Polyadenosine tails were added to capped linear transcripts using *E. coli* PolyA Polymerase (NEB) according to manufacturer's instructions, and fully processed mRNA was column purified. For circRNA, after DNase treatment additional GTP was added to a final concentration of 2 mM, and then reactions were heated at 55 °C for 15 min. RNA was then column purified. In some cases, purified RNA was recircularized: RNA was heated to 70 °C for 5 min and then immediately placed on ice for 3 min, after which GTP was added to a final concentration of 2 mM along with a buffer including magnesium (50 mM Tris-HCl, 10 mM MgCl2, 1 mM DTT, pH 7.5; New England Biolabs). RNA was then heated to 55 °C for 8 min, and then column purified. To enrich for circRNA, 20 µg of RNA was diluted in water (86 µL final volume) and then heated at 65 °C for 3 min and cooled on ice for 3 min. 20U RNase R and 10 µL of 10× RNase R buffer (Epicenter) was added, and the reaction was incubated at 37 °C for 15 min; an additional 10U RNase R was added halfway through the reaction. RNase R-digested RNA was column purified. RNA was separated on precast 2% E-gel EX agarose gels (Invitrogen) on the E-gel iBase (Invitrogen) using the E-gel EX 1–2% program; ssRNA Ladder (NEB) was used as a standard. We were unable to obtain adequate circRNA separation using other agarose gel systems. Bands were visualized using blue light transillumination and quantified using ImageJ. Unprocessed agarose gel images are present as Supplementary Fig. 8. Unprocessed agarose gel images including ladders for those gels from Figs. 1 and 2 that did not include ladders are present as Supplementary Fig. 9. For gel extractions, bands corresponding to the circRNA were excised from the gel and then extracted using a Zymoclean Gel RNA Extraction Kit (Zymogen). For high-performance liquid chromatography, 30 µg of RNA was heated at 65 °C for 3 min and then placed on ice for 3 min. RNA was run through a 4.6 × 300 mm size-exclusion column with particle size of 5 µm and pore size of 200 Å (Sepax Technologies; part number: 215980P-4630) on an Agilent 1100 Series HPLC (Agilent). RNA was run in RNase-free TE buffer (10 mM Tris, 1 mM EDTA, pH:6) at a flow rate of 0.3 mL/minute. RNA was detected by UV absorbance at 260 nm, but was collected without UV detection. Resulting RNA fractions were precipitated with 5 M ammonium acetate, resuspended in water, and then in some cases treated with RNase R as described above.

**RNase H nicking analysis**. Splicing reactions enriched for circRNA with RNase R and then column purified were heated at 65 °C for 5 min in the presence of a DNA probe (Supplementary Data 1) at five-fold molar excess, and then annealed at room temperature. Reactions were treated with RNase H (New England Biolabs) in the provided reaction buffer for 15 min at 37 C. RNA was column purified after digestion.

**Reverse transcription and cDNA synthesis**. For splice junction sequencing, splicing reactions enriched for circRNA with RNase R and then column purified were heated at 65 °C for 5 min and cooled on ice for 3 min to standardize secondary structure. Reverse transcription reactions were carried out with Superscript IV (Invitrogen) as recommended by the manufacturer using a primer specific for a region internal to the putative circRNA. PCR product for sequencing was synthesized using Q5 polymerase (New England Biolabs) and a pair of primers spanning the splice junction.

**Tissue culture and transfections**. HEK293, HEK293-GFP, HeLa, and A549 cells (ATCC) were cultured at 37 °C and 5% CO2 in Dulbecco's Modified Eagle's Medium (4500 mg/L glucose) supplemented with 10% heat-inactivated fetal bovine serum (hiFBS, Gibco) and penicillin/streptomycin. HEK293 and HeLa cells tested negative for mycoplasma. Min6 (a gift from Gordon C. Weir, Joslin Diabetes Center) medium was additionally supplemented with 5% hiFBS, 20 mM HEPES (Gibco), and 50 µM beta-mercaptoethanol (BioRad). Cells were passaged every 2–3 days. For all circRNA data sets presented in Fig. 2 except Cas9, 40–100 ng of RNase R-treated splicing reactions or HPLC-purified circRNAs were reverse transfected into 10,000 HEK293 cells/100 µL per well of a 96-well plate using Lipofectamine MessengerMax (Invitrogen) according to the manufacturer's instructions. For Cas9, 100 ng of in vitro transcribed sgRNA was reverse transfected alone or cotransfected with 150 ng of RNase R-treated Cas9 splicing reaction into 50,000 HEK293-GFP cells/500uL per well of a 24-well plate using MessengerMax. For all RNA datasets presented in Fig. 3, equimolar quantities of each RNA (equivalent to between 40 and 80 ng dependent on size) were reverse transfected into 10,000 HEK293, HeLa, or A549 cells/100 uL per well of a 96-well plate using MessengerMax. For experiments wherein protein expression was assessed at multiple time points, media was fully removed and replaced at each time point. Min6 cells were transfected in 96-well plate format between 60–80% confluency. Sample sizes were chosen based on pilot experiments to determine assay variance and to minimize reagent consumption while allowing for meaningful differences between conditions to be distinguished.

**Protein expression analysis**. For luminescence assays, cells and media were harvested 24 h post-transfection. To detect luminescence from Gaussia luciferase, 10–20 µL of tissue culture medium was transferred to a flat-bottomed white-walled plate (Corning). 25 µL of BioLux Gaussia Luciferase reagent including stabilizer (New England Biolabs) was added to each sample and luminescence was measured

on an Infinite 200Pro Microplate Reader (Tecan) after 45 s. To detect luminescence from Firefly luciferase, 100 µL of Bright-Glo Luciferase reagent (Promega) was added to each well, mixed, and incubated for 5 min. 100 µL of the culture medium and luciferase reagent mix was then transferred to a flat-bottomed white-walled plate and luminescence was detected as described above. GFP fluorescence was detected 24 h post-transfection and images were taken using an EVOS FL cell imager (Invitrogen). Erythropoietin was detected by solid phase sandwich ELISA (R&D Systems) essentially according to the manufacturer's instructions except cell culture supernatant 24 h post transfection was used, and samples were diluted 1:200 before use. Interferon-α and interleukin 6 were detected by Fireplex immunoassay (Abcam).

**Flow cytometry**. CRISPR-Cas9-mediated GFP ablation was detected by flow cytometry 96 h after transfection. HEK293-GFP and HEK293 control cells were trypsinized and suspended in Dulbecco's Modified Eagle's Medium (4500 mg/L glucose) supplemented with 10% fetal bovine serum and penicillin/streptomycin. Cells were then washed twice in FACS buffer (PBS, 5% heat-inactivated fetal bovine serum) and resuspended in FACS buffer containing Sytox Blue Dead Cell Stain (Thermo Fisher) according to the manufacturer's instructions, or FACS buffer alone for GFP and blank controls. Fluorescence was detected for 10,000 events on a BD FACSCelesta flow cytometer (BD Biosciences). Data was analyzed in Flowjo (Flowjo LLC).

**Reverse transcription and qPCR**. A total of 50,000 HeLa cells were reverse transfected with 200 ng of GLuc circRNA or modified linear mRNA. Cells were washed and RNA was harvested and purified 24 h after transfection using an RNeasy Mini Plus kit (Qiagen) according to the manufacturer's instructions. Synthesis of first-strand cDNA from total RNA was performed with High-Capacity cDNA Reverse Transcription Kit using random hexamers (Thermo Fisher Scientific). Gene specific TaqMan primers were purchased as Assay-on-Demand from (Thermo Fisher Scientific), GAPDH (Hs99999905_m1), DDX58 (Hs01061436_m1), IFN-β1 (Hs01077958_s1). The qPCR reaction was carried out using LightCycler 480 Probe Master Mix (Roche) and LightCycler 480 instrument (Roche). For each sample, the real-time PCR reaction was performed in duplicate and the averages of the obtained threshold cycle values (Ct) were processed for further calculations according to the comparative Ct method. Gene expression levels were normalized to the expression of the housekeeping gene GAPDH.

**Statistics**. Statistical analysis of the results was performed by a two-tailed unpaired Welch's *t*-test, assuming unequal variances. Differences were considered significant when $p < 0.05$. For all studies, data presented is representative of one independent experiment.

**Data availability**. The data that support the findings of this study are available from the corresponding author upon reasonable request.

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

## Acknowledgements

This work was supported in part by the Defense Advanced Research Projects Agency (Grant W32P4Q-13-1-0011). P.K. acknowledges funding from the Juvenile Diabetes Research Foundation (JDRF) postdoctoral fellowship (Grant 3-PDF-2017-383-A-N). We also thank the Nanotechnology Materials and Flow Cytometry Core Facilities at the MIT Koch Institute.

## Author contributions

R.W. and P.K. designed and executed experiments and wrote the manuscript. D.A. wrote the manuscript and provided guidance.

## Additional information

**Competing interests:** D.A., P.K., and R.W. filed a patent for the development of the circRNA technology.

