## [Peer Review File · Nature Communications]

Reviewers' comments:

Reviewer #1 (Remarks to the Author):

The paper has been improved with additional experiments and many clarifications following the referee's requests. However, the two major concerns were not appropriately addressed:

Major point 1: The authors did not check the variations in the levels of IFN-beta and RIG-I, tested in Chen et al. (2017, Molecular Cell). In this paper it is clearly shown that upon circRNA transfection the levels of RIG-I are strongly upregulated: RIG-I is induced 500-fold and this is needed to sense circRNA. This is relevant since circRNA induction of RIG-I may constitute a positive feedback loop. RIG-I may directly sense circRNA, or it may be required for a proximal signal amplification step for immune gene induction. It is important for the authors to check the level of this important mediator of interferon response upon expression of their circRNA with respect to the linear one.

To ensure comparison with their system, they should also use the same cellular lines as in Chen et al. (HeLa cells).

Major point 2: The authors used a vector able to express circRNA through autocatalytic splicing. However, they were asked to test a vector able to produce circRNA through canonical splicing and to compare the efficiency of translation. Splicing has been indeed demonstrated to be required for endogenous circRNA translation (Legnini et al., 2017, Molecular Cell). This is very important in order to compare the efficiency of the system and to sustain the hypothesis that their expression system is indeed able to provide efficient translation.

Reviewer #2 (Remarks to the Author):

The paper describes the design and cell free production of translatable circular RNAs, and based on appropriate experiments claims that the engineered circRNAs display significantly increased protein production stability compared to the unmodified or modified linear mRNAs used in this study. An important technical outcome is that circRNA production has been optimized by screening different IRES and engineering a variety of structural elements into the circRNA precursors in order to make circularization by permuted intron exon splicing most efficient. Although not being a novel method as such (production of translatable circRNA by PIE has been published previously), the technical details and the outcome of the work in terms of protein production stability are certainly of interest to others in the community and can be expected to further pave the way towards preparation of circRNAs as model compounds for structure and function studies and various applications.

The paper has been carefully edited based on Reviewers comments on the manuscript that was submitted previously to Nature Chem. Biol. The authors have carried out additional experiments, included additional references, and extended their discussion as recommended. I have just two remaining minor point to address:

1) According to the now added information on the composition of the linear mRNAs in the methods section, the modified version of the Gluc mRNA obtained from Trilink consists of "..., and complete replacement of uridine and cytosine with pseudouridine and 5-methylcytosine, respectively." I assume that only uridine and cytosine in the untranslated regions were replaced with the modified analogues? This should be precisely stated in the methods section and in the text.

2) Regarding the color code in Figure 3: It is rather hard to identify what is referred to as black or grey, because the dark and light purple in the columns is dominating the figures, rather than the lines framing the columns (in particular in Fig. 3a and c). Could this be better defined?

Reviewers' comments:

Reviewer #1 (Remarks to the Author):

The paper has been improved with additional experiments and many clarifications following the referee's requests. However, the two major concerns were not appropriately addressed:

Major point 1: *The authors did not check the variations in the levels of IFN-beta and RIG-I, tested in Chen et al. (2017, Molecular Cell). In this paper it is clearly shown that upon circRNA transfection the levels of RIG-I are strongly upregulated: RIG-I is induced 500-fold and this is needed to sense circRNA. This is relevant since circRNA induction of RIG-I may constitute a positive feedback loop. RIG-I may directly sense circRNA, or it may be required for a proximal signal amplification step for immune gene induction. It is important for the authors to check the level of this important mediator of interferon response upon expression of their circRNA with respect to the linear one.*

To ensure comparison with their system, they should also use the same cellular lines as in Chen et al. (HeLa cells).

1. We thank the reviewer for their feedback on the immunogenicity assays we have conducted. To address the direct comparison between our constructs and those used in Chen et al., we have performed an additional experiment examining the parameters requested by the reviewer. 50,000 HeLa cells were transfected with 200ng of circRNA or modified linear mRNA and RNA was harvested at 24 hours. IFN- β 1 and RIG-I transcript levels were assessed by RT-qPCR. We found that circRNA transfection led to a 30-fold induction of IFN-B mRNA and a 3-fold induction of RIG-I mRNA, as compared to mock transfection (MessengerMax), shown below and included as Supplementary Fig. 3k:

Importantly, the 3-fold induction of RIG-I that we observed is significantly lower than that reported by Chen et al. Despite the activation of IFN- β 1 and RIG-I pathways, this response was not sufficient for the shutdown of translation from transfected circRNA and did not affect the stability of expression (Fig. 3j). We acknowledge that there are differences in construct design (and therefore secondary structure) and purification methods (such as HPLC purification) used in the present manuscript.

We have amended the text to discuss these findings, as below:

“...We observed our circRNA preparations to elicit a significantly stronger IFN- α and IL-6 cytokine response in A549 cells than untransfected controls, although the response was comparable to that from unmodified linear mRNA (Supplementary Fig. 3j). We also found a 30-fold induction of IFN- β 1 and a 3-fold induction of RIG-I transcripts upon transfection of HeLa cells with circRNA, compared to mock transfection (Supplementary Fig. 3k). The 3-fold induction of RIG-I mRNA we observed was significantly lower than the 500-fold induction reported by Chen et al. in HeLa cells²⁸ and was not sufficient to suppress protein translation and the stability of protein expression from circRNA in this cellular context (Fig. 3j). The observed differences in RIG-I activation could be attributed to differences in construct design and circRNA purification methods. To further suppress an immune response directed against circRNA, methods such as nucleoside modification could potentially be incorporated into circRNA design^{13,16}....”

Major point 2: *The authors used a vector able to express circRNA through autocatalytic splicing. However, they were asked to test a vector able to produce circRNA through canonical splicing and to compare the efficiency of translation. Splicing has been indeed demonstrated to be required for endogenous circRNA translation (Legnini et al., 2017, Molecular Cell). This is very important in order to compare the efficiency of the system and to sustain the hypothesis that their expression system is indeed able to provide efficient translation.*

2. In our previous reply to the reviewer we investigated the expression of Gaussia Luciferase from transfected plasmid DNA containing the engineered splicing sequences used for in vitro RNA circularization. We compared this construct to an identical plasmid with deleted splice sites. We were unable to detect differences in expression, and therefore we were unable to verify that autocatalytic splicing was occurring in-cellulo from transfected plasmid DNA containing the optimized sequences used in this study. We therefore think that a comparison between our in-vitro splicing but in-cellulo defective autocatalytic construct and a canonical splicing construct would not yield meaningful results for the scope of this manuscript. Here we focused on showing that in-vitro transcribed and spliced circRNA is able to produce quantities of protein comparable to the in-vitro transcribed linear mRNAs used in this study, and we consider these comparisons to be appropriate and in support of the claim that in-vitro transcribed and spliced circRNA is able to provide efficient translation in comparison to in-vitro transcribed and processed linear mRNA. In the present study we do not make claims on, or intend to investigate, whether the engineered construct is able to generate efficiently translated circRNA within the context of a transfected DNA vector.

Reviewer #2 (Remarks to the Author):

The paper describes the design and cell free production of translatable circular RNAs, and based on appropriate experiments claims that the engineered circRNAs display significantly increased protein production stability compared to the unmodified or modified linear mRNAs

used in this study. An important technical outcome is that circRNA production has been optimized by screening different IRES and engineering a variety of structural elements into the circRNA precursors in order to make circularization by permuted intron exon splicing most efficient. Although not being a novel method as such (production of translatable circRNA by PIE has been published previously), the technical details and the outcome of the work in terms of protein production stability are certainly of interest to others in the community and can be expected to further pave the way towards preparation of circRNAs as model compounds for structure and function studies and various applications.

The paper has been carefully edited based on Reviewers comments on the manuscript that was submitted previously to Nature Chem. Biol. The authors have carried out additional experiments, included additional references, and extended their discussion as recommended. I have just two remaining minor point to address:

1) According to the now added information on the composition of the linear mRNAs in the methods section, the modified version of the Gluc mRNA obtained from Trilink consists of "..., and complete replacement of uridine and cytosine with pseudouridine and 5-methylcytosine, respectively." I assume that only uridine and cytosine in the untranslated regions were replaced with the modified analogues? This should be precisely stated in the methods section and in the text.

1. We have clarified that the modification of the RNAs used in this study covered the entire length of the RNA, and not just the untranslated regions, in the methods section and as below:

"...and complete replacement of uridine and cytosine along the entire mRNA with pseudouridine and 5-methylcytosine, respectively...."

2) Regarding the color code in Figure 3: It is rather hard to identify what is referred to as black or grey, because the dark and light purple in the columns is dominating the figures, rather than the lines framing the columns (in particular in Fig. 3a and c). Could this be better defined?

2. In Figures 3a and c, all of the RNAs are circular as indicated by purple bars. The black or grey outline is present to distinguish Gaussia and Firefly luciferase in 3a, and HEK293 and HeLa cell context in figure 3c. We have updated the legend of Figure 3 to clarify the assignment of colors to conditions as below:

"...Luminescence in the supernatant of HEK293 cells 24 hours after transfection with circRNA containing a panel of viral 5' UTR IRES sequences in GLuc (left axis, dark purple with black outline) and FLuc (right axis, light purple with gray outline) contexts..."

"...Luminescence in the supernatant of HEK293 (left axis, dark purple with black outline) and HeLa (right axis, light purple with gray outline) cells 24 hours after transfection with the most effective circRNAs by IRES in b)..."

REVIEWERS' COMMENTS:

Reviewer #1 (Remarks to the Author):

This reviewer has still some concern as to why the authors refuse to compare their system with that of a circRNA deriving from canonical splicing. I believe this is a crucial point for the scientific community since it will make a big difference to prepare and purify a synthetic RNA more than to produce a plasmid able to express, upon canonical splicing the wanted circRNA (that can be used also for preparing stably transfected cells). This would also help to analyze whether the observed high efficiency of translation is due to the type of IRES utilized or the construct preparation itself. Therefore, the experiment requested, namely to use their sequences in the context of canonical splicing (such as the vector ZSCANminiCirc Addgene), is still lacking.

Minor point:

The luciferase assays (Fig. 3) lacks a proper control, namely the circRNA of Gaussia/Firefly Luciferase without IRES.

Reviewer #2 (Remarks to the Author):

The request of Reviewer #1 to compare the autocatalytic system for circRNA production with that of a circRNA deriving from canonical splicing is basically comprehensible. Indeed, one could prepare stably transfected cells, and see how the expression of this canonical splice product compares to the in vitro produced circRNA.

Nevertheless, I agree with the authors that it remains questionable, whether this comparison would yield meaningful results for the scope of this manuscript. For one, cloning of the coding sequence into the 'addgene circRNA MiniVector' as recommended by Reviewer #1 and transfection constitute steps, which may harbour additional technical challenges, making comparison rather difficult. For second, expression of endogenous circRNAs, lacking the cap structure and the polyA tail, has been observed, but still is little understood (Legnini et al., 2017, Molecular Cell). Apparently, those circRNAs that are translated, contain in their UTRs IRES like elements that can drive translation. This appears to be dependent on circRNA production through a splicing event and suggests that factors loaded on the transcript upon splicing might play a crucial role in ribosome recognition and translation initiation. However, the nature of these factors and/or elements in the UTRs remains unknown. Thus, it seems more than questionable, whether production of circRNA from vector transcription and canonical splicing would deliver a translatable circRNA construct. Therefore, comparison of the circRNA construct used by the authors and the circRNA resulting from canonical splicing would be little conclusive, and I have doubts about this comparison being helpful in analysing whether the observed high efficiency of translation is due to the type of IRES utilized or the construct preparation itself, as anticipated by Reviewer #1.

In focus of the work of Anderson and coworkers is the demonstration that in-vitro transcribed and spliced circRNA is able to provide efficient translation in comparison to in-vitro transcribed and processed linear mRNA. To my opinion, this claim is well supported by the reported experiments and does not require further support by comparison of the autocatalytic system to a circRNA construct derived from canonical splicing.

REVIEWERS' COMMENTS:

Reviewer #1 (Remarks to the Author):

1. This reviewer has still some concern as to why the authors refuse to compare their system with that of a circRNA deriving from canonical splicing. I believe this is a crucial point for the scientific community since it will make a big difference to prepare and purify a synthetic RNA more than to produce a plasmid able to express, upon canonical splicing the wanted circRNA (that can be used also for preparing stably transfected cells). This would also help to analyze whether the observed high efficiency of translation is due to the type of IRES utilized or the construct preparation itself.

Therefore, the experiment requested, namely to use their sequences in the context of canonical splicing (such as the vector ZSCANminiCirc Addgene), is still lacking.

1. We thank the reviewer for their feedback on alternative methods of producing circRNA. While we agree that it would be useful to produce circRNA within cells in the context of a plasmid, we are not attempting to investigate this potential utility in the current manuscript.

2. The luciferase assays (Fig. 3) lacks a proper control, namely the circRNA of Gaussia/Firefly Luciferase without IRES.

2. We have added an additional experiment as Supplementary Fig. 4c showing that deletion of proximal or distal regions of the CVB3 IRES results in abrogated expression. While we did not delete the entire IRES, these data clearly show that the circRNA constructs we have prepared are dependent on the IRES for the translation of protein.

Reviewer #2 (Remarks to the Author):

The request of Reviewer #1 to compare the autocatalytic system for circRNA production with that of a circRNA deriving from canonical splicing is basically comprehensible. Indeed, one could prepare stably transfected cells, and see how the expression of this canonical splice product compares to the in vitro produced circRNA.

Nevertheless, I agree with the authors that it remains questionable, whether this comparison would yield meaningful results for the scope of this manuscript. For one, cloning of the coding sequence into the 'addgene circRNA MiniVector' as recommended by Reviewer #1 and transfection constitute steps, which may harbour additional technical challenges, making comparison rather difficult. For second, expression of endogenous circRNAs, lacking the cap structure and the polyA tail, has been observed, but still is little understood (Legnini et al., 2017, Molecular Cell). Apparently, those circRNAs that are translated, contain in their UTRs IRES like elements that can drive translation. This appears to be dependent on circRNA production through a splicing event and suggests that factors loaded on the transcript upon splicing might play a crucial role in ribosome recognition and translation initiation. However, the nature of these factors and/or elements in the UTRs remains unknown. Thus,

it seems more than questionable, whether production of circRNA from vector transcription and canonical splicing would deliver a translatable circRNA construct. Therefore, comparison of the circRNA construct used by the authors and the circRNA resulting from canonical splicing would be little conclusive, and I have doubts about this comparison being helpful in analysing whether the observed high efficiency of translation is due to the type of IRES utilized or the construct preparation itself, as anticipated by Reviewer #1.

In focus of the work of Anderson and coworkers is the demonstration that in-vitro transcribed and spliced circRNA is able to provide efficient translation in comparison to in-vitro transcribed and processed linear mRNA. To my opinion, this claim is well supported by the reported experiments and does not require further support by comparison of the autocatalytic system to a circRNA construct derived from canonical splicing.

1. We thank the reviewer for their feedback on the comments made by reviewer 1 and the data presented in the manuscript.